# LEAFY COTYLEDON1 expression in the endosperm enables embryo maturation in Arabidopsis

Jingpu Song [1,2,3✉], Xin Xie [1,3], Chen Chen [1,3,4], Jie Shu[1,3,4], Raj K. Thapa [1,3], Vi Nguyen[1], Shaomin Bian [1,5], Susanne E. Kohalmi [3], Frédéric Marsolais [1,3], Jitao Zou [2✉] & Yuhai Cui [1,3✉]

The endosperm provides nutrients and growth regulators to the embryo during seed development. LEAFY COTYLEDON1 (LEC1) has long been known to be essential for embryo maturation. LEC1 is expressed in both the embryo and the endosperm; however, the functional relevance of the endosperm-expressed *LEC1* for seed development is unclear. Here, we provide genetic and transgenic evidence demonstrating that endosperm-expressed *LEC1* is necessary and sufficient for embryo maturation. We show that endosperm-synthesized LEC1 is capable of orchestrating full seed maturation in the absence of embryo-expressed *LEC1*. Inversely, without *LEC1* expression in the endosperm, embryo development arrests even in the presence of functional *LEC1* alleles in the embryo. We further reveal that *LEC1* expression in the endosperm begins at the zygote stage and the LEC1 protein is then trafficked to the embryo to activate processes of seed maturation. Our findings thus establish a key role for endosperm in regulating embryo development.

[1] London Research and Development Centre, Agriculture and Agri-Food Canada, London, ON, Canada. [2] Aquatic and Crop Resource Development Research Centre, National Research Council of Canada, Saskatoon, SK, Canada. [3] Department of Biology, Western University, London, ON, Canada. [4] Molecular Analysis and Genetic Improvement Center, South China Botanical Garden, Chinese Academy of Sciences, Guangzhou, China. [5] College of Plant Science, Jilin University, Changchun, China. ✉email: jsong267@uwo.ca; jitao.zou@nrc-cnrc.gc.ca; yuhai.cui@canada.ca

Seed development in angiosperm is a complex process that is initiated by the double fertilization of egg and central cells with two sperm cells that generate the diploid embryo and the triploid endosperm, respectively[1]. The endosperm plays an essential role in seed development by nourishing the embryo via transferring maternal nutrients and growth regulators[2]. The development of embryo and endosperm depends on both parental genomes and is influenced by the communication and coordination of their genetic programs[3]. Although progress has been made[2,4], the molecular interactions between the endosperm and the embryo during seed development are generally not well understood.

Many transcription factors have been shown to regulate seed development; one of them is a nuclear factor Y (NF-Y) transcription factor LEAFY COTYLEDON1 (LEC1) that has been identified as a key regulator of seed development[5]. In the embryo, LEC1 regulates seed development programs via combinatorial interactions with other transcription factors including the AFL B3 domain proteins, ABI3, FUS3, and LEC2, which are all master regulators of seed maturation[5–10]. Null mutations in LEC1 cause defective seed phenotypes, including short embryo axis, less-developed cotyledons with anthocyanin accumulation, and desiccation intolerance[11,12]. Nonetheless, the lec1 homozygous plants rescued from mutant embryos did not exhibit any morphological abnormalities; neither were developmental paces and flowering time (Supplementary Fig. 1). As previously reported[5], although there were no obvious morphological difference between wild type and the lec1-1 embryos from the globular to linear stages, embryos with purple cotyledons were observed from the lec1-1 mutant at early maturation (Supplementary Fig. 2). LEC1 expression has been detected not only in the embryo but also in the endosperm in several plant species, including Arabidopsis thaliana[13], Brassica napus[14], and soybean[10], leading to speculation about a role of LEC1 in endosperm development[5]. However, no obvious morphological defects in the endosperm have been observed in lec1 mutant seeds[15]. We hypothesized that the endosperm-expressed LEC1 may act as a molecular signal in the early communication between endosperm and embryo, and subsequently exerts its key roles in activating and regulating various embryo developmental programs. In the following sessions, we present the results of our genetic and transgenic experiments designed to test this hypothesis.

## Results

**Expression of LEC1 in the endosperm is required for seed maturation.** To investigate if the expression of LEC1 in the endosperm was necessary for seed maturation, we generated seeds with unfertilized endosperms of lec1 genotype and fertilized diploid embryos of LEC1 genotype using a genomic imprinting bypassing strategy (Fig. 1a), through which small but fully developed seeds could be produced when fis-class mutant flowers were crossed with cdka;1 mutant pollens[16,17]. We first generated two double mutant lines, fis2-6$^{+/-}$ lec1-1$^{-/-}$ and cdka;1$^{+/-}$ lec1-1$^{-/-}$. In the self-crossed siliques of the double mutants, half of the seeds were arrested at early development stages, as was seen in the self-crossed fis2-6$^{+/-}$ and cdka;1$^{+/-}$ single mutants. The rest of the sibling seeds showed lec1 mutant phenotype as expected (Fig. 1b and Supplementary Fig. 3a). To obtain seeds with lec1 endosperms and LEC1 embryos, we crossed fis2-6$^{+/-}$ lec1-1$^{-/-}$ flowers with cdka;1$^{+/-}$ pollens (Fig. 1a). Meanwhile, crosses of fis2-6$^{+/-}$ × cdka;1$^{+/-}$ and fis2-6$^{+/-}$ × cdka;1$^{+/-}$ lec1-1$^{-/-}$ were also conducted as positive controls, and that of fis2-6$^{+/-}$ lec1-1$^{-/-}$ × cdka;1$^{+/-}$ lec1-1$^{-/-}$ as negative control (Supplementary Fig. 3b–d). The F1 siliques had three types of seeds: normal size (CDKA;1$^{+/+}$), small (cdka;1$^{+/-}$), and aborted (Fig. 1b–m). The normal sized seeds and small seeds were classified as not aborted

(NA) (Fig. 1c) and they were distinguished by the size of seed area (normal seeds: above 0.15 mm$^2$, small seeds: <0.1 mm$^2$) as reported previously[16]. Of note, genotype CDKA;1$^{+/+}$ represents the normal sized seeds (Fig. 1d–g) which were wild type for both the FIS2 and CDKA;1 genes (maternal FIS2-6$^+$ CDKA;1$^+$; paternal FIS2-6$^+$ CDKA;1$^+$) since the maternal FIS2-6$^+$ CDKA;1$^+$ paternal FIS2-6$^+$ cdka;1$^-$ genotype seeds aborted at the pre-globular stage (Fig. 1m). Genotype cdka;1$^{+/-}$ represents the small seeds (Fig. 1h–k) which were formed from maternal fis2-6$^-$ CDKA;1$^+$ and paternal FIS2-6$^+$ cdka;1$^-$ since the maternal fis2-6$^-$ CDKA;1$^+$ paternal FIS2-6$^+$ CDKA;1$^+$ genotype seeds aborted at the heart stage (Fig. 1l). Further, the genotypes of the CDKA;1$^{+/+}$ and cdka;1$^{+/-}$ seeds were confirmed by PCR-based genotyping at the CDKA;1$^+$/cdka;1$^-$ and the LEC1$^+$/lec1$^-$ loci in the seedlings derived from each type of seeds (Supplementary Fig. 3e). Notably, the small seeds from the positive controls contained fully developed embryos (Fig. 1h, i). The small seeds from the crosses of fis2-6$^{+/-}$ lec1-1$^{-/-}$ × cdka;1$^{+/-}$, with lec1$^{-/-}$ endosperm and LEC1$^{+/-}$ embryo, on the other hand, showed defective embryo phenotype (Fig. 1j) that was also observed in the small seeds from the negative control, with lec1$^{-/-}$ endosperm and lec1$^{-/-}$ embryo (Fig. 1k). Together, these results indicated that the expression of LEC1 in the endosperm was necessary for embryo maturation.

**Haploid seeds with LEC1 endosperms and lec1 embryos develop normally.** To test whether the expression of LEC1 in the embryo was required for seed maturation, we generated seeds with haploid lec1 embryos and normal LEC1 endosperms by pollinating flowers of a haploid induction line (SeedGFP-HI), which was generated by introducing a transgene expressing an altered form of CENH3 fused with GFP and also a transgene expressing GFP driven by the seed storage protein gene At2S3 promoter in the cenh3-1 line[18,19], with lec1-1 pollen grains (Fig. 2a). We also conducted crosses of SeedGFP-HI flowers with wild type (WT) pollens to serve as control. From the SeedGFP-HI × lec1-1 and SeedGFP-HI × WT crosses, we harvested the mature siliques containing a mixture of haploids and hybrid diploids in addition to a high frequency of aborted seeds (Fig. 2b). The diploid and haploid seeds were further hand-sorted based on their fluorescence patterns, as described previously[18]. The diploids with both embryo and endosperm inheriting the At2S3: GFP transgene in their maternal genome from the maternal parent SeedGFP-HI developed into seeds with uniform GFP signal, while the haploids with only the endosperm inheriting the maternal genome developed into seeds with mottled GFP fluorescence (Fig. 2c). From the crosses of SeedGFP-HI × lec1-1, haploid seeds with no lec1-1 seed defect (i.e., dark purple color) were obtained, which phenotypically resembled the haploid seeds from the SeedGFP-HI × WT crosses (Fig. 2c). To examine the embryo phenotypes, we dissected the diploid and haploid seeds. We observed that both diploid and haploid seeds collected from the SeedGFP-HI × lec1-1 and the SeedGFP-HI × WT crosses contained fully developed embryos. The phenotypes of those diploid and haploid embryos highly resembled WT embryos and were clearly distinguishable from the lec1-1 embryos (Fig. 2d). Diploid embryos were confirmed with the presence of GFP fluorescence and haploid embryos were GFP-negative (Fig. 2d).

The dissected diploids and haploids embryos were placed on MS agar plates to develop for subsequent genotype analysis. The diploid embryos were identifiable based on GFP signals observed at the centromeres in the root tip cells (Fig. 2e), because they inherited the CENH3-GFP transgene from the HI line. The haploid embryos only inherited genome information from the paternal plant lec1-1, thus displaying no GFP signal (Fig. 2e). To further validate the identities of the diploids and haploids, we examined the morphology and genotype of the plants developed

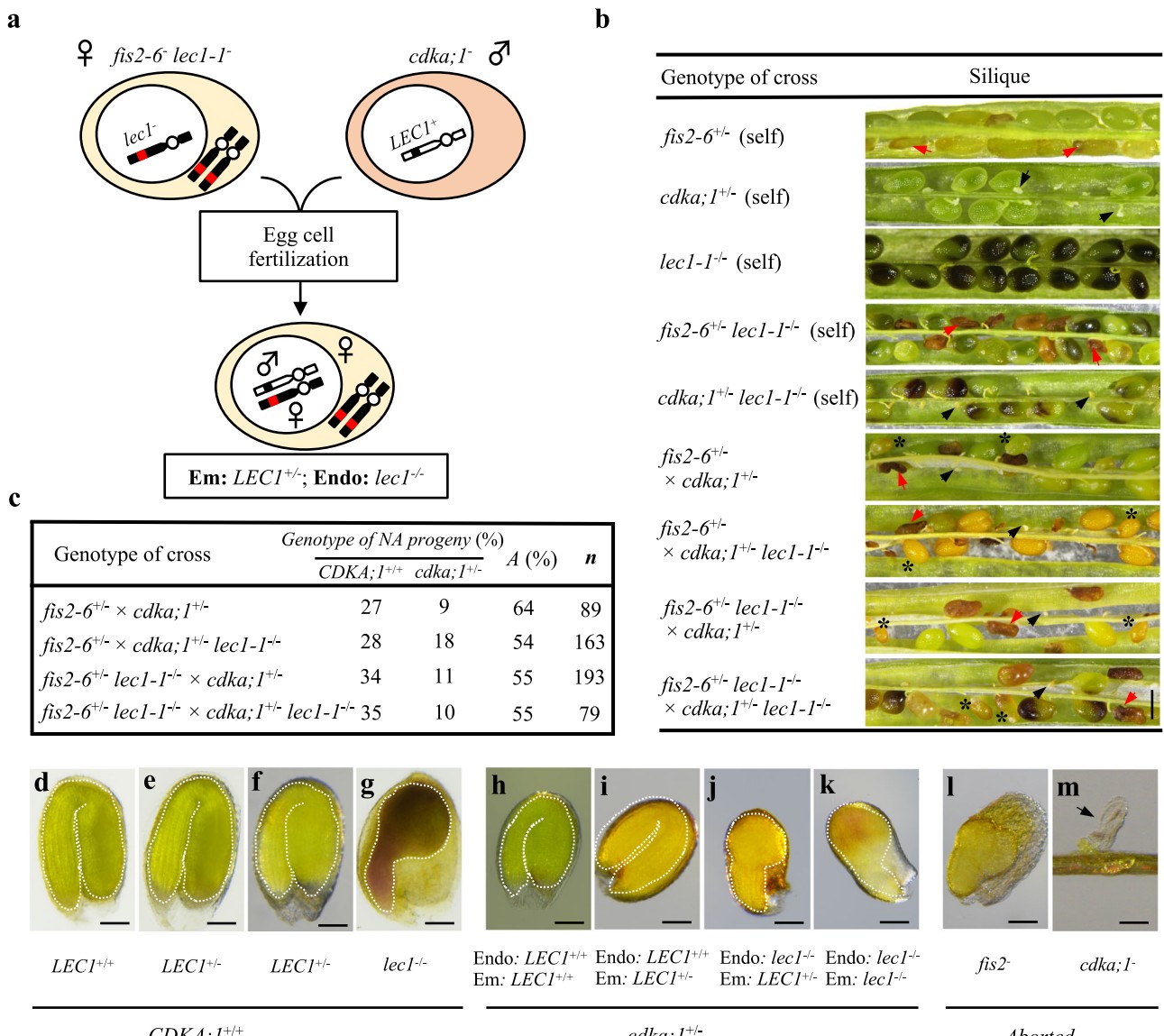

**Fig. 1 Loss-of-function of *LEC1* in the endosperm disrupts seed development. a** Cartoon showing a *fis2-6⁻ lec1⁻* female gamete crossing with *a cdka;1⁻ LEC1⁺* pollen to generate a small seed with *lec1⁻/⁻* endosperm and *LEC1⁺/⁻* embryo. Red dot, mutant *lec1⁻* allele; black dot, wild type *LEC1⁺* allele. Em embryo, Endo endosperm. **b** Green mature siliques collected from different genetic crosses. *lec1* seeds display purple color. Arrows represent aborted seeds (red, *fis2⁻* phenotype; black, *cdka;1⁻* phenotype); asterisks indicate small seeds generated from the *fis2-6⁻ × cdka;1⁻* bypassing cross. Five siliques from each of the five crosses were examined. Scale bar: 500 μm. **c** Percentage of various progeny produced from the bypassing genetic crosses. *CDKA;1⁺/⁺* indicate the normal sized wild type seeds, while *cdka;1⁺/⁻* indicate the small seeds. NA not aborted seeds, A aborted seeds, n number of seeds scored. **d–k** Images of the *CDKA;1⁺/⁺* (normal size) and the *cdka;1⁺/⁻* (small) seeds with different genotypes produced from the bypassing genetic crosses. Five seeds from each genotype were examined. The crosses of *fis2-6⁺/⁻ × cdka;1⁺/⁻* produce **d** type normal seeds and **h** type small seeds. The crosses of *fis2-6⁺/⁻ × cdka;1⁺/⁻ lec1-1⁻/⁻* produce **e** type normal seeds and **i** type small seeds. The crosses of *fis2-6⁺/⁻ lec1-1⁻/⁻ × cdka;1⁺/⁻* produce **f** type normal seeds and **j** type small seeds, while the crosses of *fis2-6⁺/⁻ lec1-1⁻/⁻ × cdka;1⁺/⁻ lec1-1⁻/⁻* produce **g** type normal size *lec1* seeds and **k** type small seeds. **l** An example of aborted seeds produced by *fis2-6⁻*. Five seeds were examined. **m** An example of aborted seeds produced by *cdka; 1⁻*. Five seeds were examined. Scale bar: 100 μm.

from the dissected embryos. In comparison with the diploid plants, the haploid plants showed reduced stature featuring narrower and smaller leaves (Supplementary Fig. 4a), as previously observed[19]. During the reproductive phase, the diploid plants derived from the SeedGFP-HI × *lec1-1* crosses produced normal size siliques that were filled with seeds segregating for the *lec1-1* and wild type phenotypes (Supplementary Fig. 4b). In contrast, the haploid plants from the same crosses produced only a few siliques with 1–2 seeds of the *lec1-1* genotype. Similarly, haploid plants from the SeedGFP-HI × WT crosses produced siliques containing 1–2 seeds of the wild type genotype

(Supplementary Fig. 4b). To confirm that the haploids from the SeedGFP-HI × *lec1-1* cross only inherited the genome from the *lec1-1* paternal parent, we genotyped both the diploid and haploid plants derived from the cross at the *LEC1* locus and detected both the wild type *LEC1* allele and the *lec1-1* T-DNA allele in the diploid plants but only the *lec1-1* T-DNA allele in the haploid plants (Supplementary Fig. 4c). These results further confirmed the identity of the haploid and the diploid seeds generated from the SeedGFP-HI × *lec1-1* crosses.

The facts that the presence of wild type *LEC1* allele inherited from the maternal parent in the endosperm of haploids as

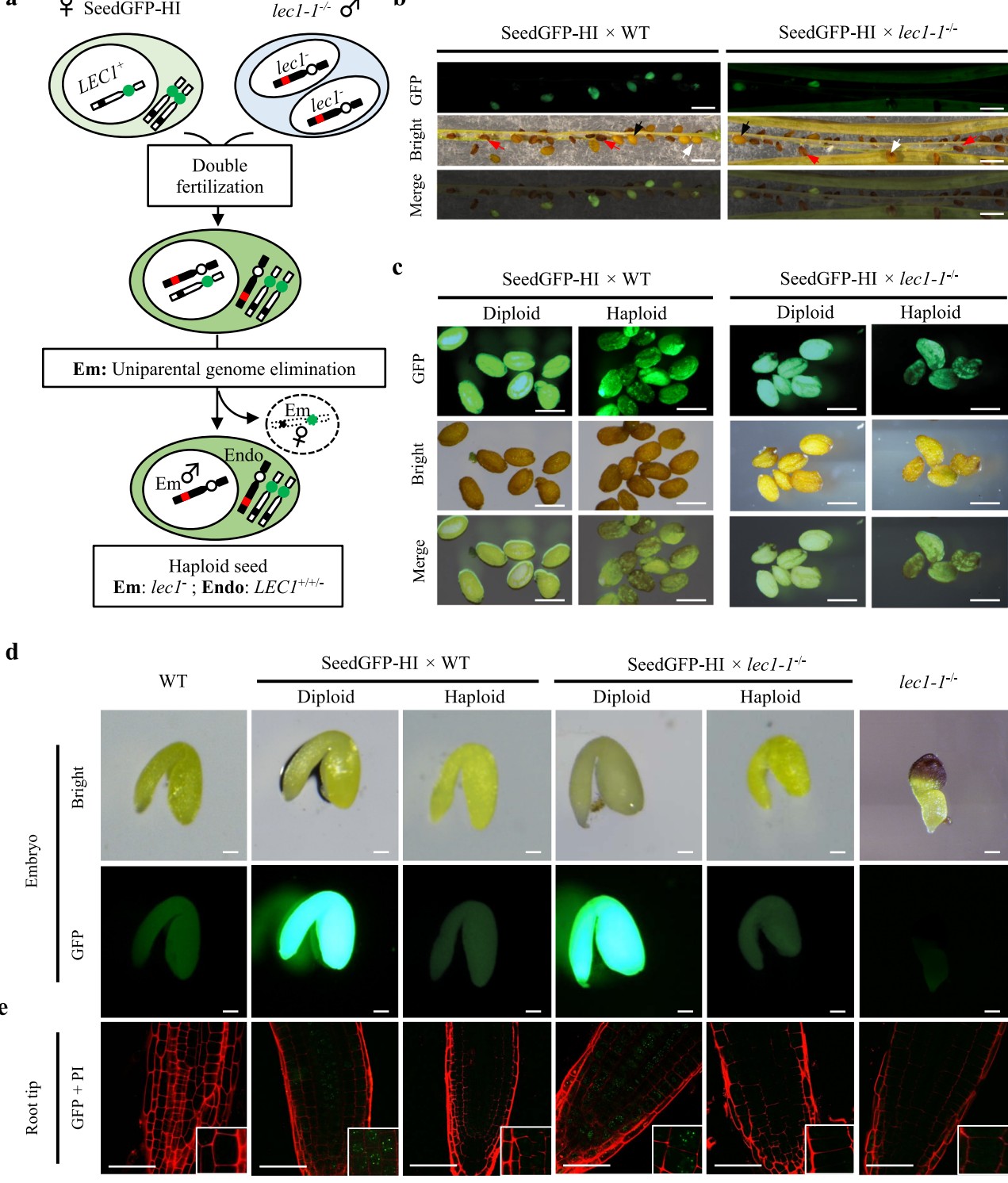

evidenced by the mottled seed GFP signals and that the haploid seeds did not exhibit any *lec1-1* defective seed phenotypes such as purple cotyledons and short embryo axis (Fig. 2c, d) indicate that the seeds with *LEC1* endosperm and *lec1* embryo could develop normally through maturation. These results demonstrated that *LEC1* gene expression in the embryo was dispensable for embryo maturation.

**Expressing *LEC1* exclusively in the endosperm rescues the *lec1* seed phenotype.** To further investigate whether an exclusive

expression of *LEC1* in the endosperm can rescue *lec1* seed defects, we employed two endosperm-specific promoters, *proPHERES1* (*pPHE*)[20,21] and *proZHOUPI* (*pZOU*)[22], to direct the expression of *LEC1* in *lec1* mutants. We built two endosperm-specific expression constructs, *pPHE::LEC1-GFP* (PPL) and *pZOU::LEC1-GFP* (PZL), as well as controls that include *pLEC1::LEC1-GFP* (PLL), *pPHE::ABI3-GFP* (PPA), *pZOU::ABI3-GFP* (PZA), *pPHE:: GFP* (PPg), and *pZOU::GFP* (PZg) (Supplementary Fig. 5). These constructs were introduced individually into embryo-rescued *lec1-3* homozygous plants. From the T1 generation of PPL and

**Fig. 2 Fully developed haploid seeds with *lec1* embryos and *LEC1* endosperms. a** Cartoon showing the genetic cross using the SeedGFP-HI line as female parent and *lec1-1* as male parent to induce haploid progeny. Red dot: mutant *lec1⁻* allele; black dot: wild type *LEC1⁺* allele; green dot: weak centromeres CENH3-GFP from the HI parent; white dot: wild type centromeres. Em embryo, Endo endosperm. Note: the *At2S3:GFP* reporter in the SeedGFP-HI line is not shown in the cartoon. **b** Typical siliques containing GFP florescent seeds produced from the SeedGFP-HI × wild type (WT) or SeedGFP-HI × *lec1-1* crosses. Approximately 30 siliques were produced from each type of the genetic crosses. Red arrows represent aborted seeds; black arrows indicate diploid seeds; white arrows indicate haploid seeds. Scale bar: 1000 μm. **c** Diploid and haploid seeds hand-sorted by their GFP signal patterns: seeds with uniform GFP signals (diploid) and seeds with mottled GFP fluorescence (haploid). At least four seeds from each genotype were examined for each replicate. This experiment was repeated three times independently. Scale bar: 500 μm. **d** Hand-dissected embryos from the mature diploid and haploid seeds produced from the crosses. Seed GFP florescence were detected under UV lights. Three embryos from each genotype were examined. Scale bar: 100 μm. **e** Detection of GFP signals in the centromere of root tip cells under confocal microscope. Insets: magnified areas of root cells indicated. Cell walls stained with propidium iodide (PI) are shown in red. Confocal images are shown as the merged channel of GFP and PI. Three embryos of each seed type were examined. This experiment was repeated three times independently. WT and *lec1-1* embryos were used as control. Scale bar: 50 μm.

PZL plants, 67% and 74% of the seeds (T2), respectively, were rescued to normal phenotype (Fig. 3a). Among the lines introduced with the control constructs, 79% of the embryos from the PLL plants showed normal phenotype, but none were found in the PPA, PPg, and PZA lines (Fig. 3a). We then tested the germination of T2 seeds to test if the transgenic seeds were desiccation-tolerant. As shown in Fig. 3b and Supplementary Fig. 6, seeds from the PPL, PZL, and PLL lines germinated successfully, while those from PPA, PPg, and PZA succumbed to desiccation. Morphologically, the mature seeds of PPL, PZL, and PLL were indiscernible from that of wild type, but the PPA, PPg, and PZA seeds resembled that of *lec1-3* (Fig. 3c–m). The PPL, PZL, and PLL seeds possessed wild type levels of storage proteins (12S and 2S) while PPA, PPg, and PZA exhibited much lower levels similar to that of *lec1-3* (Fig. 3n). In addition, the same transgene constructs were also introduced into the *lec1-1* plants in parallel and analogous results were obtained (Supplementary Fig. 7). Hence, *LEC1* directed to be expressed solely in the endosperm was fully capable of complementing *lec1* mutant seed in embryo morphology, seed germination, and storage protein accumulation. These transgenic experiments, in combination with the above described genetic evidence obtained from the haploid seed experiment, established that exclusive expression of *LEC1* in the endosperm is sufficient to capacitate normal embryo development.

**The onset of *LEC1* expression occurs first in the endosperm**. To decipher the initiation and the mode of action of the endosperm-expressed *LEC1*, we closely monitored GFP signals in the developing seeds of PLL, PPL, and PPg from the zygote stage onward, to maturation stage (Fig. 4a–u). Pollen grains and ovules in the PLL before fertilization were also examined, and no GFP signal could be detected (Supplementary Fig. 8a, b). GFP signals were first emerged from fertilized central cell (endosperm) nuclei in the PLL, PPL, and PPg seeds at the zygote stage (Fig. 4a–c and Supplementary Fig. 8c–e). By the two-cell stage, GFP signals began to appear in the pro-embryo of PLL, in both the endosperm and the pro-embryo of PPL, but restricted to endosperm only in the PPg seeds (Fig. 4d–f and Supplementary Fig. 8f–h). In the PLL embryos, GFP signals were observed from the globular to bent stages, but not at the maturation stage (Fig. 4g, j, m, p, s). Similarly, the PPL embryos exhibited GFP signals at the globular, heart, linear, and bent stages, but not the maturation stage (Fig. 4h, k, n, q, t). Notably, strong signals were present in the suspensors of PLL and PPL embryos (Fig. 4d, e, g, h, j, k). Similar to observations from the PPL, GFP signals were also detected in developing embryos of the PZL seeds (Supplementary Fig. 8i, j). In contrast, there was no GFP signal in the PPg embryos at any stages (Fig. 4c, f, i, l, o, r, u). In the PPA seed, consistent with its *lec1* mutant embryo phenotype, GFP signals were only detected in the endosperm nuclei at the pre-globular stage, but not in the embryos (Supplementary Fig. 9).

To determine whether the LEC1 in the embryos of PPL was mobilized from the endosperm, we further generated a PPL-GFP₃ line expressing LEC1 fused to 3 copies of GFP (Supplementary Fig. 10a). Such fusion proteins with multimeric GFPs have been employed successfully by others to restrict the mobility of mobile transcription factors[23,24]. LEC1-GFP₃ signals were observed in the endosperm nuclei at the pre-globular stage, but not in the embryos at the pre-globular, globular, heart, or linear stages (Supplementary Fig. 10b). These results suggest that the LEC1 fusion protein was restricted to the endosperm in the PPL-GFP₃ seeds. Consistent with the lost mobility of LEC1-GFP₃, the fusion gene failed to rescue the *lec1-1* seed defects (Supplementary Fig. 10c).

In addition, we generated another control, the *pAtML1::LEC1-GFP lec1-1⁻/⁻* (PML) line, employing the embryo-specific *AtML1* promoter to drive the exclusive expression of LEC1 in the embryo[25]. As expected, expression of the LEC1-GFP fusion in the PML was only observed in the embryos, and it could successfully rescue the *lec1-1* seed defects (Supplementary Fig. 11). This result is consistent with earlier observations that the *lec1* mutant endosperms do not show any obvious defect[5], and is also in line with our observations presented above (Supplementary Fig. 2).

In light of the endosperm specificity of *pPHE*, these results signified that in the PPL seeds the endosperm-expressed LEC1-GFP was trafficked to the embryo from its expression origin, the endosperm, to enable embryo maturation. Such an scenario is further supported by our data from the PPL-GFP₃ line. These results show that LEC1 was mobilized from the endosperm to the embryo at very early stages of seed development.

**The LEC1 protein enters the embryo from the endosperm**. To investigate whether LEC1 was mobilized from the endosperm to the embryo in the form of RNA or protein, we performed RT-qPCR analyses to determine if there was any *LEC1-GFP* mRNA in the embryos. First, we measured the relative expression of *GFP* in the homozygous PLL, PPL, PPg, and WT (negative control) whole seeds at the linear stage, confirming the presence of *GFP* transcripts in the PLL, PPL, and PPg seeds (Fig. 4v). We then measured the *GFP* mRNA in the linear stage embryos of PLL, PPL, using PPg and WT as negative controls since they had no GFP signal in the embryos. *GFP* mRNA was detected in the PLL embryos but not in the PPL embryos (Fig. 4w). This result, when combined with the observation of clear presence of LEC1-GFP signals in the PPL embryos, strongly suggests that LEC1 was transported in the form of protein from the endosperm to the embryo. Such a notion is also consistent with our observation that there was no GFP signal in the PPL-GFP₃ embryos.

We lastly examined the effectiveness of endosperm-synthesized LEC1 in activating embryo maturation genes known to be

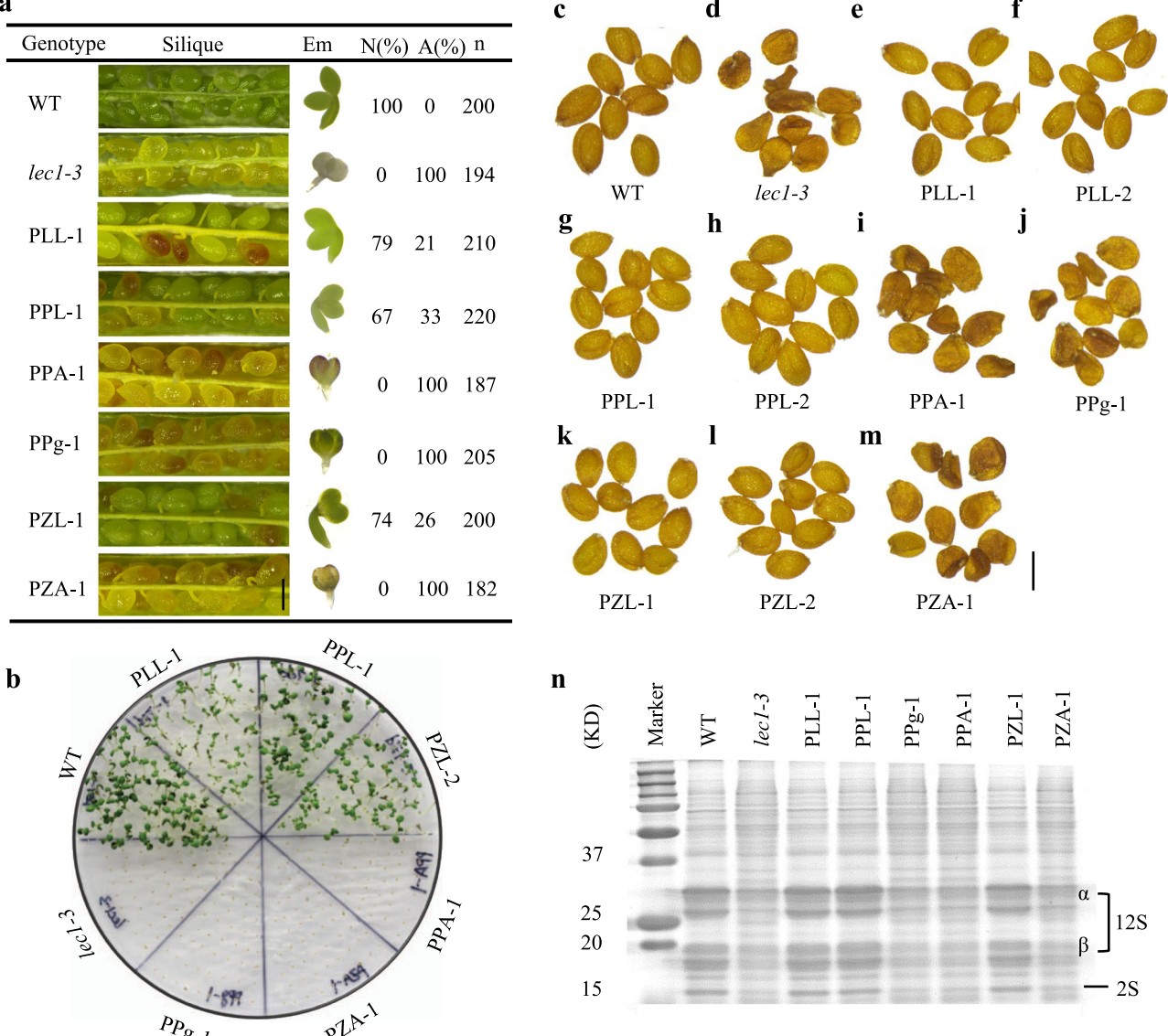

**Fig. 3 Gain-of-function of *LEC1* in the endosperm rescues *lec1-3* mutant seed phenotype. a** Green mature siliques collected from T1 plants with different transgenes in the *lec1-3* background. Siliques of T1 plants PLL-1 (*pLEC1::LEC1-GFP lec1-3⁻/⁻*), PPL-1 (*pPHE::LEC1-GFP lec1-3⁻/⁻*), and PZL-1 (*pZOU::LEC1-GFP lec1-3⁻/⁻*) contain two types of seeds: [WT] seeds and [lec1-3] seeds. PPA-1 (*pPHE::ABI3-GFP lec1-3⁻/⁻*), PPg-1 (*pPHE::GFP lec1-3⁻/⁻*), and PZA-1 (*pZOU:: ABI3-GFP lec1-3⁻/⁻*) only produce defective seeds (*lec1-3*). [WT] indicates WT phenotype, [lec1-3] indicates *lec1-3* phenotype. At least five independent individual transgenic lines were examined for each construct transformation. Images of a typical embryo from each line are shown on the right next to the images of the siliques. WT (Ler-0) and *lec1-3* are used as control. N normal seeds, A abnormal seeds (*lec1-3*), n number of seeds scored. Em embryo. Scale bar: 500 μm. **b** Germination of seeds with different transgenic backgrounds at day 7. Fifty seeds were used for one biological germination test. This experiment was repeated three times independently. **c-m** Phenotypes of mature seeds from each of the transgenic backgrounds as indicated. Ten seeds from each genotype were examined. Scale bar: 300 μm. **n** SDS-PAGE gel image showing the 2S and 12S storage proteins in seeds from each of the transgenic backgrounds. This experiment was repeated three times independently.

downstream target genes of LEC1[5,26], including the AFL B3 transcription factor genes *ABI3*, *FUS3*, and *LEC2*. Through analyzing a set of LEC1 ChIP-seq data from a previous study[10], we were able to verify the LEC1 occupancy on the promoters of *LEC2*, *FUS3*, *ABI3* and *LEC1* (Supplementary Fig. 12). We then measured the relative expression levels of *LEC2*, *FUS3*, and *ABI3* in the linear stage embryos of PLL, PPL, and PPg. As shown in Fig. 4x, similar levels of *LEC2*, *FUS3*, and *ABI3* expression was detected in the PPL and PLL embryos, but almost negligible in the PPg embryos. These results demonstrated that the LEC1 protein synthesized in the endosperm adequately activates the seed development programs including the maturation genes without de novo LEC1 synthesis in the PPL embryo.

## Discussion

Our study uncovers a mode of action of *LEC1* that is initially expressed in the endosperm but subsequently in the form of LEC1 protein mobilized to the embryo as a molecular signal (Fig. 5). Given the LEC1-GFP strong signals being detected in the suspensors, we conjecture that the route of trafficking is via the suspensor, which has long been suspected of mediating symplastic transfer of metabolites and proteins from the endosperm to embryo[2]. The endosperm-synthesized LEC1 protein, once transported into the embryo, can subsequently trigger the expression of *LEC1* in the embryo and ultimately the *LEC1*-regulatory network to enable normal seed development. Such an scenario is also consistent with the postulated activity of LEC1 as

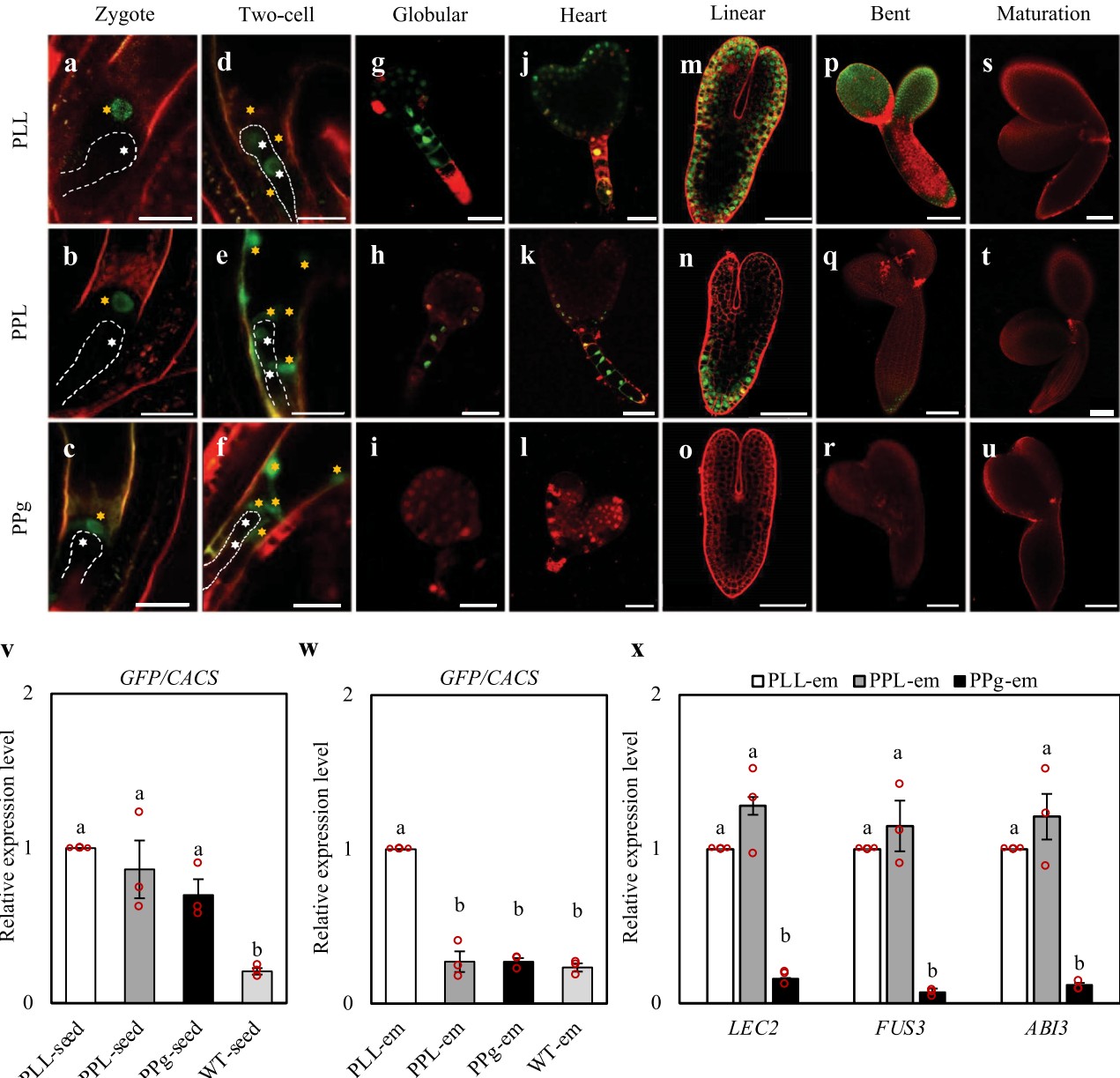

**Fig. 4 Endosperm-expressed LEC1 enters embryo to regulate seed maturation genes. a–u** Localization patterns of GFP signals in PLL, PPL, and PPg seeds at various seed development stages from zygote to maturation as indicated. Five seeds or embryos at each stage from each line were examined. Embryos (**a–f**) are outlined in white dash lines for clarity. Amber stars indicate endosperm nuclei and white stars indicate embryo nuclei. GFP signals are shown in green. Cell walls stained with PI are shown in red. Shown here are merged confocal images from GFP and PI channels. Scale bars: 20 μm (**a–l**); 50 μm (**m–o**); 100 μm (**p–u**). **v** Relative expression of *GFP* in the whole seeds of PLL, PPL, PPg, and WT at linear stage. **w** Relative expression of *GFP* in the embryos (em) of PLL, PPL, PPg, and WT seeds at linear stage. **x** Relative expression of *LEC2, FUS3* and *ABI3* in the embryos (em) of PLL, PPL, PPg seeds at linear stage. **v–x** a to b indicate statistical difference with one-way ANOVA followed by the post-hoc Tukey multiple comparison tests (*p* < 0.05). The *CACS* gene was used as an internal control. Values are mean ± standard error of three biological replicates.

a pioneer transcription factor[27] and the notion that the chromatin environment in the endosperm is distinct from that in the embryo[3]. Seed genes, particularly those pertinent to seed maturation, are believed to be repressed during vegetative growth by the polycomb proteins-mediated chromatin condensation[28–30]. How they get reset during seed development has been a puzzling question. Our findings provide a plausible explanation for the reprograming of these genes in the new generation (i.e., seed). It is conceivable that, due to the favorable chromatin and cellular environments in the endosperm, *LEC1* becomes activated there shortly after fertilization and enters the embryo, where it overcomes the repressive chromatin context to trigger the de-repression of seed genes including itself.

Why would a functional *LEC1* copy remain expressed in the embryo, given that its expression in the endosperm seems to be enough to ensure proper seed development? It would be tempting to speculate that one advantage would be to have more-than-enough LEC1, as an enhanced mechanism, to enable/double-secure the normal embryo development, a vital process for a plant's survival. Another puzzling question is why LEC1 expression in the embryo is normally dependent on LEC1 protein provided to the embryo by the endosperm. We would speculate that such an arrangement allows a spatial separation of the embryo and its key regulator, thus adding a new layer of control in the regulation of embryo development. This would make sure

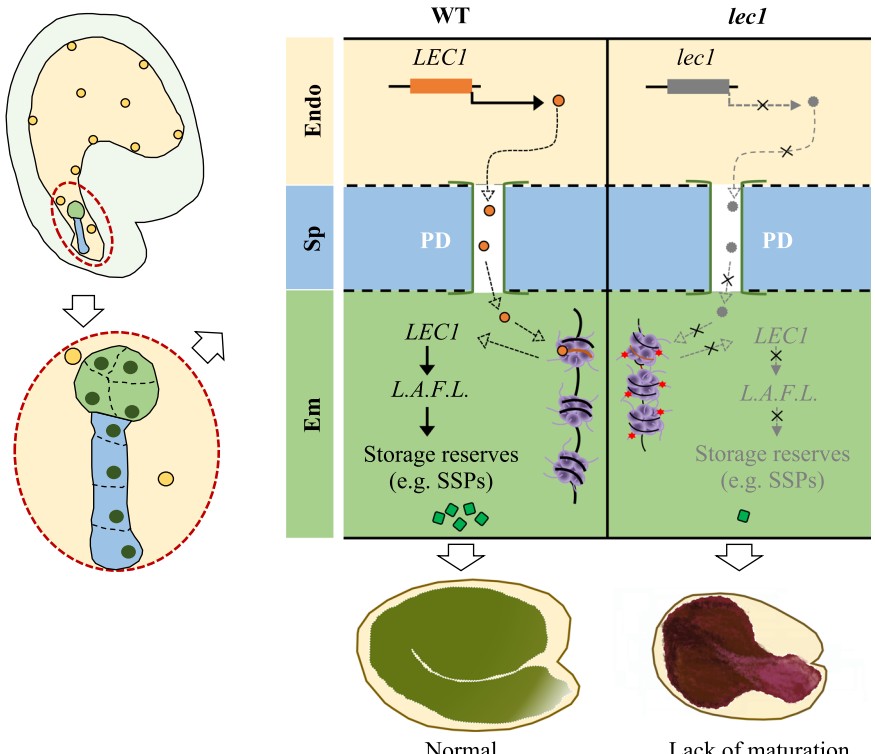

**Fig. 5 A model depicting the regulation of seed maturation by the mobile transcription factor LEC1.** In early stages of seed development, the onset of *LEC1* expression occurs in the endosperm, followed by mobilization of the LEC1 protein to the embryo, likely via the plasmodesmata (PD) in the suspensor cells. After entering the embryo, the endosperm-synthesized LEC1 overcomes the repressive chromatin constraints to trigger the de-repression of seed maturation-related transcription factor genes including *LEC2, ABI3, FUS3,* and *LEC1* itself (*L.A.F.L.*). Subsequently, the L.A.F.L. transcription factors activate the expression of seed storage reserve synthesis genes, enabling normal seed maturation. In the *lec1* mutant seeds, the L.A.F.L. network remains repressed due to lack of *LEC1* expression in both the endosperm and the embryo, thus leading to dramatically reduced accumulation of seed storage reserves and consequently the formation of desiccation-intolerant purple seeds. Notes: endosperm (Endo), embryo (Em), and suspensor (Sp) are colored with light amber, light green, and light blue, respectively; amber circles represent endosperm nuclei; dark green circles indicate embryo nuclei; orange and gray circles represent wild type and mutant LEC1 proteins, respectively; black lines represent genomic DNA and the orange/black rectangular bars indicate the wild type and mutant *LEC1* genes, respectively; purple circles represent histones; red stars represent repressive histone modifications, such as H3K27me3; and bright green squares represent seed storage reserves (SSPs: seed storage proteins).

that embryo development, in particular the maturation program, would not initiate until it is informed to do so, i.e., after successful fertilization, otherwise remains securely repressed during the rest of the plant's life cycle.

In conclusion, we demonstrate the critical importance of endosperm-synthesized LEC1 in enabling normal seed development including maturation in *Arabidopsis*. These findings establish LEC1 as a molecular signal in the communication between the embryo and the endosperm during seed development, and thus provide novel insights into the mechanisms by which the endosperm nourishes and regulates embryo development.

## Methods

**Plant materials and growth conditions**. *Arabidopsis thaliana* ecotypes Columbia-0 (Col-0) and Landsberg (Ler-0) were used as wild types. *Arabidopsis* mutant lines *lec1-1* (SALK_131219)[27], *lec1-2* (CS870475), *lec1-3* (CS5739)[31], *cdka;1* (SALK_106809.34.90.X)[16,17], and *fis2-6* (CS6998) were obtained from the Arabidopsis Biological Resource Center (ABRC). Haploid Inducer SeedGFP-HI line was a gift from UC Davis[18,19]. Transgenic plants of *pLEC1::LEC1-GFP lec1* (PLL), *pPHE::LEC1-GFP lec1* (PPL), *pPHE::GFP lec1* (PPg), *pPHE::ABI3-GFP lec1* (PPA), *pZOU::LEC1-GFP lec1* (PZL), *pZOU::GFP lec1* (PZg), *pZOU::ABI3-GFP lec1* (PZA), *pPHE::LEC1-GFP-GFP-GFP lec1* (PPL-GFP₃), and *pAtML1::LEC1-GFP lec1* (PML) were generated in this work. Note: *lec1* can be either *lec1-1* or *lec1-3*, as specified in the main text. Double mutant lines *lec1-1 cdka;1* and *lec1-1 fis2-6* were produced by genetic crosses. The wild type and mutant seeds were germinated either in soil or on half-strength MS medium and grown in a growth room with humidity of 65% under a 16 h light/8 h dark cycle at 22 °C. All genotypes were determined by PCR,

by resistance to hygromycin, or by phenotype. Primers used for genotyping are listed in Supplementary Table 1.

**Plasmid construction**. To generate the *pLEC1::LEC1-GFP* construct, the *LEC1* genomic region from 2000 bp upstream of the ATG to the end of ORF without stop codon was amplified from Col-0 genomic DNA with primers *F-pro-LEC1/R-LEC1*, transferred into pMDC107[32]. To generate the *pPHE::LEC1-GFP* construct, the *LEC1* genomic region from ATG to the end of ORF without stop codon was amplified from Col-0 genomic DNA with primers *F-LEC1/R-LEC*, ligated into pMDC107 to get the *LEC1-GFP* construct. Then the *PHE* promoter region (2000 bp upstream of the ATG) was amplified from Col-0 DNA with primers *F-pro-PHE/R-pro-PHE*, transferred to the *LEC1-GFP* construct. To build the *pPHE::LEC1-GFP-GFP-GFP* construct, *GFP-GFP* was amplified with primers *Pme1-Mlu1-GFP-F/Asc1-GFP-R*, transferred to pMDC107 to get the *GFP-GFP-GFP* construct. Then *pPHE1::LEC1* was cloned with primers *F-pro-PHE/Mlu1-LEC1-R*, then transferred to the *GFP-GFP-GFP* construct. Similarly, to get the construct *pPHE::ABI3-GFP*, the *ABI3* genomic region from ATG to the end of ORF without stop codon was amplified from Col-0 genomic DNA with primers *F-ABI3/R-ABI3*, then ligated into pMDC107 to get the *ABI3-GFP* construct before introducing the *PHE* promoter amplified previously. To get the *pPHE::GFP* construct, the *PHE* promoter region (2000 bp upstream of the ATG) was amplified from Col-0 DNA with primers *F-pro-PHE/R-pro-PHE-1*, ligated into pMDC107. To generate the *pZOU::LEC1-GFP* construct, the *ZOU* promoter region (2000 bp upstream of the ATG) was amplified from Col-0 DNA with primers *F-pro-ZOU/R-pro-ZOU*, then ligated to the *LEC1-GFP* construct. To get the construct *pZOU::ABI3-GFP*, the *ZOU* promoter amplified previously was cloned into *ABI3-GFP*. To get the *pZOU::GFP* construct, the *ZOU* promoter region (2000 bp upstream of the ATG) was amplified from Col-0 DNA with primers *F-pro-ZOU/R-pro-ZOU-1*, ligated into pMDC107. To generate the *pAtML1::LEC1-GFP* construct, the *AtML1* promoter region[25] was amplified from Col-0 DNA with primers *pAtML1-F/pAtML1-R*, then ligated to the *LEC1-GFP* construct. At least five independent transgenic lines for each construct

transformation were obtained and examined. Primer information is listed in Supplementary Table 1. Plant transformation was performed via floral dipping[33].

**Seed rescue**. Immature siliques from *lec1* heterozygous parental plants were collected and surface sterilized with 70% ethanol, dissected under a dissecting microscope. Homozygous mutant seeds (purple color, with defective embryos) were transferred to half-strength MS agar plates with 1% sucrose for germination. After germination, seedlings were then transferred to soil for further experiments. *lec1* homozygosity was normally confirmed by PCR-based genotyping.

**Genetic crosses for the *fis2-cdka;1* bypassing assay and haploid seed production**. To obtain the *lec1-1$^{-/-}$ fis2-6$^{+/-}$* line, the *lec1-1* homozygous mutant and *fis2-6* heterozygous mutant plants were used for cross. The *lec1-1$^{-/-}$* flowers at stage 12 were emasculated and shortly after that the pistils were hand-pollinated with pollen grains from the *fis2-6$^{+/-}$* plants. F1 seeds from the cross were collected and planted to produce F2 seeds. The *lec1-1$^{-/-}$ fis2-6$^{+/-}$* plants were selected based on the combination of phenotypes of *lec1-1$^{-/-}$* (purple color, not fully developed embryos) and *fis2-6$^{+/-}$* (aborted seeds at heart stage) from the F2 self-crossed siliques. Similarly, the *lec1-1$^{-/-}$ cdka;1$^{+/-}$* line was obtained by crossing the *lec1-1$^{-/-}$* line and the *cdka;1$^{+/-}$* line. The *cdka;1$^{+/-}$* flowers at stage 12 were emasculated and the pistils were hand-pollinated with pollen grains from the *lec1-1$^{-/-}$* plants. F1 seeds from the crossed siliques were used to produce F2 seeds. The *lec1-1$^{-/-}$ cdka;1$^{+/-}$* plants were confirmed by the combination of phenotypes of *lec1-1$^{-/-}$* (purple color, not fully developed embryo) and *cdka;1$^{+/-}$* (half amount of seeds aborted at heart stage in one silique) from the F2 self-crossed siliques. For FIS-CDKA bypassing assays, *fis2-6$^{+/-}$* and *lec1-1$^{-/-}$ fis2-6$^{+/-}$* lines were used as maternal parents while *cdka;1$^{+/-}$* and *lec1-1$^{-/-}$ cdka;1$^{+/-}$* plants were used as pollen donor. For SeedGFP-HI experiments, the SeedGFP-HI flowers were emasculated at stage 12 and hand-pollinated with pollen grains from the *lec1-1$^{-/-}$* plants. Seeds were collected from the F1 siliques for further experiments.

**Microscopy**. For examination of seed morphology, siliques were hand-dissected for differential interference contrast (DIC) imaging by using a stereomicroscope. For examination of embryo development, seeds were mounted in the Hoyer's solution for DIC[34]. The GFP florescence of seeds and embryos produced from the HI crosses were examined under UV lights with a stereomicroscope. The stereomicroscope used for the experiments was a Nikon SM225 equipped with a DS-Ri2 camera (Nikon). Confocal images were taken using an Olympus Fluoview FV1200 laser scanning microscope with excitation wavelength of 488 nm for EGFP and 559 nm for propidium iodide (PI). Confocal images were analyzed using the imaging software: OLYMPUS FLUOVIEW Ver.4.2.

**RT-qPCR analysis**. For RNA isolation, linear embryos were dissected as described previously with some modifications[35]. More specifically, individual embryos were hand-dissected under a stereomicroscope (EMZ PLS-2 stand, MEIJI) from seeds immersed in 10% RNAlater (Thermo Fisher Scientific) with fine point tweezers, and transferred with a 2–20 μl RNAse-free pipette tip (VWR) to depression slides (VWR) containing 200 μl of 10% RNAlater. After every 10 embryos dissected, the embryos were washed with 10% RNAlater three times, then transferred to 30 μl of 100% RNAlater. RNA was isolated from each pool of 50 embryos per sample by adding 500 μl TRIzol (Thermo Fisher Scientific) followed by incubation at 60 °C for 30 min, and then purified according to the TRIzol reagent protocol for RNA isolation from small quantities of tissue (Life Technologies). For each sample, 100 ng of RNA was used in reverse transcription reactions using a iScript Reverse Transcription Supermix for RT-qPCR kit (Bio-RAD). For each quantification-PCR (qPCR), SsoFast EvaGreen Supermix (Bio-RAD) with Gene-specific and CACS (endogenous control)[36] primers were used to conduct qPCR reactions on a Bio-RAD C1000™ Thermal Cycler with the CFX96™ Real-time PCR System (Bio-RAD). qPCR data was analyzed using Bio-Rad CFX Manager program. Primer information is listed in Supplementary Table 1.

**Storage protein analysis**. Mature seeds were ground in the extraction buffer (100 mM Tris-HCl pH 8.0, 1% SDS, 10% glycerol, and 2% β-mercaptoethanol). The extracts were boiled for 3 min, followed by centrifugation at 20,000 *g* for 5 min. The supernatant of each sample was transferred to a new tube. Protein samples were mixed with 5× loading buffer, denatured by adding 18.5 mM dithiothreitol, before loading onto 15% SDS-PAGE gels. After separation by electrophoresis using a Biochrom Novaspec Plus Visible Spectrophotometer (Bio-RAD), the protein gels were stained with Coomassie Brilliant Blue R250 for 30 min, followed by de-staining for 1 h with de-staining solution (10% glacial acid, 40% methanol) before imaging with a GS-900 Calibrated Densitometer scanner.

**Statistical analysis**. R v.3.6.3 (http://www.r-project.org/) was used for the statistical analyses. Bartlett tests were performed to verify the equality of the variance across the samples. One-way ANNOVA analyses and post-hoc Tukey tests were conducted to determine the significant difference. Bar graphs were generated by using Microsoft Excel 2016.

**Reporting summary**. Further information on research design is available in the Nature Research Reporting Summary linked to this article.

## Data availability
All lines used in the study will be provided upon signature of appropriate material transfer agreement. All data are available in the main text or the supplementary materials. Source data are provided with this paper.

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

## Acknowledgements

We thank the Arabidopsis Biological Resource Center for providing the mutant seeds used in this study, and Dr. Maruthachalam Ravi and Dr. Mohan Marimuthu (University of California, Davis) for kindly providing us the SeedGFP-HI seeds. This work was supported by grants from the Natural Science and Engineering Research Council of Canada (RGPIN/04625-2017 to Y.C.), Agriculture and Agri-Food Canada (to Y.C.), and the Sustainable Food System program of Aquatic and Crop Resource Development Research Centre, National Research Council of Canada (to J.Z.).

## Author contributions

J.Z. and Y.C. conceived the project; C.C., Y.C., J. Song, and J.Z. designed the experiments; J. Song conducted most of the experiments; X.X. generated the PPL-GFP$_3$ and PML lines, propagated transgenic lines, and conducted genotyping. V.N. performed the storage protein extraction experiment; J. Shu and R.T. contributed to the genetic work and molecular analyses; S.B., S.K., and F.M. contributed to data analysis and supervision; J. Song, J.Z., and Y.C. wrote the manuscript.

## Competing interests

The authors declare no competing interests.
