## [Peer Review File · Nature Communications]

REVIEWER COMMENTS

Reviewer #1 (Remarks to the Author):

In this paper the authors report the functional characterization of LEC1, especially questioning whether the expression of the gene in the endosperm or in the embryo fulfills the function. The main claim is that endosperm-expressed LEC1 is capable of controlling seed maturation in the absence of embryo expressed LEC1, in Arabidopsis. This control would imply protein trafficking from the endosperm to the embryo. The paper contains main 4 figures, 9 supplementary figures and 1 additional table.

The article is short and concise, well written with high quality and informative figures. The question of how the control of the expression is made in young embryos after fertilization has occurred between the different organs (endosperm, zygote, testa) is a hot topic with important implications for seed development and reserve accumulation.

The article uses nice genetic resources and mutants to specify expression of genes and reporter genes in specific territories of the developing seed. The methods are however very short.

Of note, the methods for microscopy are not informative enough and several pictures are claimed to be fluorescence microscopy although they strongly resemble confocal microscopy (figure sup 8c, d, e, f, g, h, Figure 2 d, e, Figure 4 a, d, g, j, b, e, h, k, c, f, I, l). Please give filter sets used, wave length, type of illumination, camera. If a confocal microscope was used, lasers, detectors, brand etc.... On the same line, methods on statistical analysis should be reinforced (ANOVA parameters, verification of the equality of the variance across samples with a Bartlett test for example, type of excel package used to perform ANOVA as standard Excel is rather poor for statistical analyses).

1- The entire demonstration is based on the use of a very tricky but nice genetic system based on *fis cdk* mutants to express LEC1 in the endosperm or in the embryo. Thus the embryo is never "normal" as claimed line 61, figure 1 as the embryo is only hemizygous for LEC1 and dosage effects cannot be excluded, altering the reading of the phenotypes.

2- I suggest to have on figure 1 a clear example of *lec1* phenotype such as extended figure 7 a for non LEC1 specialists (i.e. a row of purple seeds).

3- On extended fig 7, b *lec1-1* phenotype is presented but is weaker than c-PPg (less purple), which is not normal. On figure 7c the phenotypes are as severe between the genotypes.

4- Extended figure 1: I do not fully understand the use of *lec2* here. It is not used in the main text and is not helping to the demonstration.

5- Avoid references in the abstract (line 15, 18)

6- In the abstract line 27: I disagree with the claim "uncover a novel mode of communication between the endosperm and the embryo". There is a demonstration of communication but the authors do not give indication about novelty of the communication nor on the implemented communication. There is apparent protein trafficking.

7- Figure extended 6b: please provide statistical analysis between germination rates

8- Figure 4: here authors present a GFP signal in h and k and compare it with autofluorescence in i and l but channels are missing: you should have autofluorescence for PPL and GFP for PPg (even with no signal) to be able to compare. Same for PLL -p GFP signal compared to PPL-q and PPg -r (autofluorescence): a row should be added presenting GFP in bent cotyledons and another with autofluorescence in bent cotyledons for PLL, PPL and PPg. Last, LEC1 expression in bent cotyledons is strange (PPL-p) with no nuclear localization and an expression pattern really different from already published. This should be carefully checked with spectrum analysis to ensure this expression is not due to autofluorescence. As the wave lengths of the filters have not been provided, this makes reading difficult.

9- Line 166: LEC1 occupancy on LEC2, FUS3 and ABI3 promoters. Please specify what is the "input" on the figure 9, indicate the TSS, the tool used to determine this TSS. It is written "promoter regions" but for LEC2 and ABI3, I do not see a peak under only the promoter region but also in the coding region. Please explain. What are the negative controls to ensure LEC1 is not binding to all promoters...?

10- If the authors are right, LEC1 is accumulated in the endosperm, migrates to the embryo where it plays its role of pioneer factor, including activating its own copy in the embryo. But as expression in the endosperm only seems to be enough to ensure proper seed development and complement *lec1*, why would a functional copy remain expressed in the embryo? I would recommend to provide an additional figure with a proposed operating model for LEC1

expression/accumulation in the endosperm and embryo for the discussion.

Reviewer #2 (Remarks to the Author):

LEC1 is a well-known regulator of embryogenesis and the seed maturation program. The work described here employs sophisticated genetic manipulations to support a rather bold and surprising claim: that LEC1 expression in the endosperm is required as well as sufficient for providing LEC1 function to the entire seed. The claim rests in large part on the finding that embryos developing in a *lec1* mutant endosperm show a mutant phenotype even if their genome contains a wild type LEC1 allele; and that conversely embryos developing in a LEC1 endosperm show no obvious defects even if they are mutant for *lec1*. The expression of LEC1-GFP fusions from endosperm-specific promoters corroborate these findings and further suggests that LEC1 protein may move from the endosperm through the suspensor into the embryo.

This is a very intriguing study, and I enjoyed reading it. There seems to be a good chance that some of the author's findings will spark controversy. Thus, it seems important to address a number of weaknesses that may (potentially significantly) limit the value of these experiments.

Creation of LEC1 embryos in *lec1* mutant endosperm by bypassing endosperm imprinting: The design of Figure 1c (and the Extended Data Figure 3) is copied from Table 1 of the original paper on "bypassing genomic imprinting" (Novack & al., 2007). There, the table reports transmission data for several alleles used in the study, and "Genotype of viable progeny (%)" indeed refers to F1 plants genotyped by PCR for the respective mutations. In contrast, both Figure 1c and Extended Data Figure 3 show phenotypic classes. Genotypes are inferred from the observed phenotypes, and it would much improve the clarity of presentation if the authors explained in the text how this was done. As far as I can tell, the "Aborted" category (which doesn't really contribute to the "viable progeny") refers to either aborted ovules (fertilization defects are very common with *cdk* pollen, consistent with phenotypes marked by black arrowheads in Figure 1b) or aborted seeds (maternal *FIS2* paternal *cdk* are predicted to cause a pre-globular arrest; maternal *fis* paternal *CDK* are predicted to result in an arrest at about the heart, consistent with phenotypes in marked by red arrowheads in Figure 1b). The remaining two classes are referred to as *CDK+/+* and *cdk+/-* and represent "large seeds" (*CDK+/+*, which are also maternal *FIE* since the maternal *FIE* paternal *cdk* seeds are predicted to abort) and "small seeds" (*cdk+/-*, which are also maternal *fie*, since the maternal *fie* paternal *CDK* seeds are predicted to abort). The expected frequency of these classes is not discussed in the paper, probably since the authors make a qualitative argument: "small seed" resulting from crosses of *fis/+ lec1* pistils to *cdk/+* pollen (predicted genotype: maternal *fie*; paternal *cdk*; endosperm *lec1/lec1*; embryo *LEC1/lec1*) contain abnormal embryos, suggesting that LEC1 activity in the endosperm is required for normal development of the embryo. This is a surprising result. A weakness of the experiment is that only a few select phenotypes are shown in detail (Figure 1 d-g), although panel b suggests quite a bit of variability, and that the genotypes of viable embryos are not directly assessed ("small" vs. "large" seed does not seem like stringent enough criteria in this case). The frequency of genotypes segregating in the viable progeny of key crosses should be directly determined, even if the desiccation intolerance of *lec1* seeds complicates the task; in addition, the phenotypes of the corresponding seeds/embryos should be documented (as in Figure 1 d-e) to better assess the overall variability and the correlation of genotype and phenotype.

Creation of *lec1* embryos in normal endosperm by genome reduction: While the genotype of the haploid plants produced by crossing with *lec1* pollen has been documented with care, the authors provide very little information on the phenotype of haploid embryos/seeds (one image is included in Figure 2) or even on the numbers of haploid embryos/plants that were examined as part of the experiment. Typically, the seeds produced by haploid inducer lines show a wide range of abnormalities, and it seems important to document the phenotypic spectrum of the haploid and hybrid embryos/seeds produced in this particular cross to ensure the analysis is representative. The phenotypic spectrum of mutant embryos/seeds produced by *lec1* or *lec1/+* plants should be documented side-by-side to enable a direct comparison. Finally: while I agree with the authors

that the endosperm of seeds containing haploid (*lec1*) embryos is unlikely to have completely lost its wild type *LEC1* copy, this has not been shown.

Analysis of *LEC1*-GFP movement: Fusions to multimeric GFPs are typically included in similar studies as non-mobile or less mobile controls. Such controls should be included here. Another control that may be considered is expression of *LEC1* from an embryo-specific promoter, a promoter that does not confer expression in the endosperm. Expression of *FUS3* with an *ML1* promoter has been reported to fully complement *fus3* mutant embryos (Tsuchiya, 2004), and it would be interesting to see what activities an analogous *LEC1* construct has.

Figure 4: A reporter with a nuclear GFP would be a more appropriate control for showing that the *PHERES* promoter is not active in the embryo; the GFP of *pMDC107* and hence *PPg* contains no nuclear localization signal (although, strangely, localization in c & f seems nuclear), and it seems problematic to compare cytoplasmic signals to the nuclear signals of *PPL*.

The figure legend does not specify which channels are shown in the various panels – from the looks of it, the rows showing “Globular”, “Heart”, “Bent” and “Maturation” embryos show only a single channel: cyan (autofluorescence) or green (GFP), which wouldn’t make much sense.

I don’t understand why the *PPg-m* sample is used as a control in panel w. Detection by RT-PCR is likely much more sensitive than detection by fluorescence. A non-transgenic sample would be the appropriate control, showing the noise produced in the reaction; in addition, it would enable testing the expectation that the *PHERES* promoter does not confer expression in embryos.

Discussion: It is well-documented that *LEC1* is transcribed in the developing embryo until the maturation phase. Thus, the perhaps the most puzzling implication of this work is that *LEC1* expression in the very domain that requires *LEC1* activity cannot provide the activity. The authors should comment on this implication in their discussion.

Response to Reviewers' Comments

We are grateful to the reviewers for their insightful and constructive suggestions. Reviewer #1 critiqued on the lack of details in the microscopy and statistical analysis. Reviewer #2 commented positively on our experimental design and results but suggested some further experiments to improve our manuscript. We have since conducted new experiments and analyses as suggested. The suggestions have proved to be very helpful in improving the manuscript. Our detailed point-to-point responses to the reviewer's specific comments are presented below:

REVIEWER COMMENTS

Reviewer #1 (Remarks to the Author):*In this paper the authors report the functional characterization of LEC1, especially questioning whether the expression of the gene in the endosperm or in the embryo to fulfils the function. The main claim is that endosperm-expressed LEC1 is capable of controlling seed maturation in the absence of embryo expressed LEC1, in Arabidopsis. This control would imply protein trafficking from the endosperm to the embryo. The paper counts main 4 figures, 9 supplementary figures and 1 additional table*
The article is short and concise, well written with high quality and informative figures. The question of how the control of the expression is made in young embryos after fertilization has occurred between the different organs (endosperm, zygote, testa) is a hot topic with important implications for seed development and reserve accumulation.
The article uses nicely genetic resources and mutants to specify expression of genes and reporter genes in specific territories of the developing seed. The methods are however very short. Of note, the methods for microscopy is not informative enough and several pictures are claimed to be fluorescence microscopy although they strongly resemble confocal microscopy (figure sup 8c, d, e, f, g, h, Figure 2 d, e, Figure 4 a, d, g, j, b, e, h, k, c, f, I, l). Please give filter sets used, wave length, type of illumination, camera. If a confocal microscope was used, lasers, detectors, brand etc.....

Response: We thank the reviewer for this suggestion. In this revised version, we have provided detailed information in the Methods section; see lines 319-325.

On the same line, methods on statistical analysis should be reinforced (Annova parameters, verification of the equality of the variance across samples with a Bartlett test for example, type of excel package used to perform Annova as standard Excel is rather poor for statistical analyses).

Response: The statistical analysis has been reinforced using R v.3.6.3; please see lines 353-356.

1. The entire demonstration is based on the use of a very tricky but nice genetic system based on fis cdk mutants to express LEC1 in the endosperm or in the embryo. Thus the embryo is never “normal” has claimed line 61, figure 1 as the embryo is only hemizygous for LEC1 and dosage effects cannot be excluded, altering the reading of the phenotypes.

Response: We agree with the reviewer the term “normal” was not precise in context and we have replaced it with “fertilized diploid” to indicate the fact that in terms of genome ploidy the embryo is “normal”, in contrast to the abnormal genome ploidy level in the endosperm (2n, instead of 3n); please see line 57. It has been shown in previous reports and also in this work that heterozygous LEC1/*lec1* seeds do not show any seed defect during development. These observations suggest that LEC1 is a dominant gene and lack of only one LEC1 allele does not have dosage effect on the seed phenotype.

*2. I suggest to have on figure 1 a clear example of *lec1* phenotype such as extended figure 7 a for non LEC1 specialists (i.e. a row of purple seeds).*

Response: We appreciate the reviewer’s excellent suggestion and we have added a clear example of *lec1* phenotype in Fig.1b. Please see the revised Fig.1b.

*3. On extended fig 7, b *lec1-1* phenotype is presented but is weaker than c-PPg (less purple), which is not normal. On figure 7c the phenotypes are as severe between the genotypes.*

Response: In Supplementary Fig.7b, the mature seeds were cleared with Hoyer’s solution for better imaging. Due to the longer exposure to the Hoyer’s solution, the purple colour in the seeds could have been faded. Indeed, the *lec1-1* seeds show individual difference with a range of purple colour in the same silique. We have repeated this experiment with more care and provided the results in the new version. Please see the revised Supplementary Fig.7b.

4. *Extended figure 1: I do not fully understand the use of lec2 here. It is not used in the main text and is not helping to the demonstration.*

Response: We fully agree with this comment and have excluded the use of *lec2* in the manuscript. Please see the revised Supplementary Fig.1.

5. *Avoid references in the abstract (line 15, 18).*

Response: This has been corrected now.

6. *In the abstract line 27: I disagree with the claim “uncover a novel mode of communication between the endosperm and the embryo”. There is a demonstration of communication but the authors do not give indication about novelty of the communication nor on the implemented communication. There is apparent protein trafficking.*

Response: We appreciate the reviewer’s insight and agree with the comment. We have revised it to “Our findings thus establish a key role for endosperm in regulating embryo development.”; please see lines 24-25.

7. *Figure extended 6b: please provide statistical analysis between germination rates.*

Response: As suggested, we have provided statistical analysis. Please see the revised Supplementary Fig. 6b and legend.

8. *Figure 4: here authors present a GFP signal in h and k and compare it with autofluorescence in i and l but channels are missing: 1). you should have autofluorescence for PPL and GFP for PPG (even with no signal) to be able to compare. Same for PLL -p GFP signal compared to PPL-q and PPg -r (autofluorescence): a row should be added presenting GFP in bent cotyledons and another with autofluorescence in bent cotyledons for PLL, PPL and PPg. 2). Last, LEC1 expression in bent cotyledons is strange (PPL-p) with no nuclear localization and an expression pattern really different from already published. This should be carefully checked with spectrum analysis to ensure this expression is not due to autofluorescence. 3). As the wave lengths of the filters have not been provided, this makes reading difficult.*

Response: We thank the reviewer for providing detailed suggestions. We have now provided: 1) image with overlapped channel of GFP and PI to demonstrate our results as shown in the revised Fig.4. 2). Yes, LEC1 expression pattern was revealed previously. Based on the review from Jo, et al. (2019), LEC1 expression could be detected from the pre-globular to the bent stage, but was not seen in the maturation stage. Thus, it makes sense to see GFP signals in the bent stage of PLL embryo (the reviewer wrote PPL-p; we assumed the reviewer rather meant PLL-p). 3). We have provided the detailed information for our confocal microscopy in the Methods, please see lines 322-325.

9. *Line 166: LEC1 occupancy on LEC2, FUS3 and ABI3 promoters. Please specify what is the “input” on the figure 9, indicate the TSS, the tool used to determine this TSS. It is written “promoter regions” but for LEC2 and ABI3, I do not see a peak under only the promoter region but also in the coding region. Please explain. What are the negative controls to ensure LEC1 is not binding to all promoters...?*

Response: Genomic input (no antibody) was used as “input”, as described in Pelletier et al. (2017). The promoter regions are defined as the 2kb genomic regions upstream of TSS (transcription start site). The determination of TSSs were based on the annotation of the Arabidopsis genome, which are based on either transcript sequencing in most cases or prediction in some other cases. Yes, over the LEC2 and ABI3 loci, LEC1 not only binds to the promoter regions, but also to the gene bodies. We don't know the functional relevance of the LEC1 occupancy over the gene bodies yet, but such binding patterns have also been observed for other transcription regulators. We have included a negative control (data at *PHE1* is presented to show no LEC1 binding). Please see the revised Supplementary Fig.12 and legend.

10. *If the authors are right, LEC1 is accumulated in the endosperm, migrates to the embryo where it plays its role of pioneer factor, including activating its own copy in the embryo. But as expression in the endosperm only seems to be enough to ensure proper seed development and complement *lec1*, why would a functional copy remain expressed in the embryo? I would recommend to provide an additional figure with a proposed operating model for LEC1 expression/accumulation in the endosperm and embryo for the discussion.*

Response: We greatly appreciate the reviewer for the insightful suggestion A proposed model has been added as Fig.5 in the revised manuscript. While we do not have a precise answer to the question of why would a functional copy remain expressed in the embryo, we speculate that one advantage would be to have more-than-enough LEC1, as an enhanced mechanism, to enable/double-secure the normal embryo development, a vital process for plants' survival. Please see lines 234-244.

Reviewer #2 (Remarks to the Author):

LEC1 is a well-known regulator of embryogenesis and the seed maturation program. The work described here employs sophisticated genetic manipulations to support a rather bold and surprising claim: that LEC1 expression in the endosperm is required as well as sufficient for providing LEC1 function to the entire seed. The claim rests in large part on the finding that embryos developing in a lec1 mutant endosperm show a mutant phenotype even if their genome contains a wild type LEC1 allele; and that conversely embryos developing in a LEC1 endosperm show no obvious defects even if they are mutant for lec1. The expression of LEC1-GFP fusions from endosperm-specific promoters corroborate these findings and further suggests that LEC1 protein may move from the endosperm through the suspensor into the embryo.

This is a very intriguing study, and I enjoyed reading it. There seems to be a good chance that some of the author's findings will spark controversy. Thus, it seems important to address a number of weaknesses that may (potentially significantly) limit the value of these experiments.

Creation of LEC1 embryos in lec1 mutant endosperm by bypassing endosperm imprinting:

The design of Figure 1c (and the Extended Data Figure 3) is copied from Table 1 of the original paper on "bypassing genomic imprinting" (Novack & al., 2007). There, the table reports transmission data for several alleles used in the study, and "Genotype of viable progeny (%)" indeed refers to F1 plants genotyped by PCR for the respective mutations. In contrast, both Figure 1c and Extended Data Figure 3 show phenotypic classes. Genotypes are inferred from the observed phenotypes, and it would much improve the clarity of presentation if the authors explained in the text how this was done.

Response: We acknowledge this point and have modified the text accordingly. We have changed "genotype of viable progeny" to "genotype of NA (Not Aborted) progeny". Please see revised Fig. 1c. We have provided the detailed description of how we classified the seed phenotypes. Please see lines 67 -79.

As far as I can tell, the "Aborted" category (which doesn't really contribute to the "viable progeny") refers to either aborted ovules (fertilization defects are very common with cdk pollen, consistent with phenotypes marked by black arrowheads in Figure 1b) or aborted seeds

(maternal *FIS2* paternal *cdk* are predicted to cause a pre-globular arrest; maternal *fis* paternal *CDK* are predicted to result in an arrest at about the heart, consistent with phenotypes in marked by red arrowheads in Figure 1b). The remaining two classes are referred to as *CDK*^{+/+} and *cdk*^{+/-} and represent “large seeds” (*CDK*^{+/+}, which are also maternal *FIE* since the maternal *FIE* paternal *cdk* seeds are predicted to abort) and “small seeds” (*cdk*^{+/-}, which are also maternal *fie*, since the maternal *fie* paternal *CDK* seeds are predicted to abort). The expected frequency of these classes is not discussed in the paper, probably since the authors make a qualitative argument: “small seed” resulting from crosses of *fis*^{+/+} *lec1* pistils to *cdk*^{+/+} pollen (predicted genotype: maternal *fie*; paternal *cdk*; endosperm *lec1/lec1*; embryo *LEC1/lec1*) contain abnormal embryos, suggesting that *LEC1* activity in the endosperm is required for normal development of the embryo. This is a surprising result. A weakness of the experiment is that only a few select phenotypes are shown in detail (Figure 1 d-g), although panel b suggests quite a bit of variability, and that the genotypes of viable embryos are not directly assessed (“small” vs. “large” seed does not seem like stringent enough criteria in this case). The frequency of genotypes segregating in the viable progeny of key crosses should be directly determined, even if the desiccation intolerance of *lec1* seeds complicates the task; in addition, the phenotypes of the corresponding seeds/embryos should be documented (as in Figure 1 d-e) to better assess the overall variability and the correlation of genotype and phenotype.

Response: We thank the reviewer for the helpful criticism and valuable suggestions. We have conducted genotyping, for both the *cdka;1* and *lec1* loci, on the “normal” and “small” seeds (young seedlings were used, which represent the genotype of embryos; for each seed type, three independent seeds/seedlings were genotyped) (please see lines 76-79) and provided the genotyping results in the revised Supplementary Fig. 3e. In the revised manuscript and figures, we have documented all the phenotypes of the corresponding seeds produced from the crosses including normal seeds as well as the aborted seeds including the seeds arrested at pre-globular stage (maternal *FIS2*, paternal *cdka;1*) and the seeds arrested at heart stage (maternal *fis2*, paternal *CDKA;1*). Please see the revised Fig. 1d-g and 1l-m. We also assessed the frequency of genotypes segregating in the NA progeny of crosses “*fis2-6*^{+/-} × *cdka;1*^{+/-}”, “*fis2-6*^{+/-} × *cdka;1*^{+/-} *lec1-1*^{-/-}”, “*fis2-6*^{+/-} *lec1-1*^{-/-} × *cdka;1*^{+/-}”, and “*fis2-6*^{+/-} *lec1-1*^{-/-} × *cdka;1*^{+/-} *lec1-1*^{-/-}” (Fig. 1c).

Our data showed that the ratio of “normal” and “small” seeds was 2~3 : 1. In theory, the ratio of “normal” to “small” seeds should be 1:1 as the result of genotype segregating. In the original study by Nowack, et. al. (2007), they also observed a similar ratio and thought that the fertilization bias was due to the paternal effect conferred by the *cdka;1* pollen.

*Creation of *lec1* embryos in normal endosperm by genome reduction:*

*While the genotype of the haploid plants produced by crossing with *lec1* pollen has been documented with care, the authors provide very little information on the phenotype of haploid embryos/seeds (one image is included in Figure 2) or even on the numbers of haploid embryos/plants that were examined as part of the experiment. Typically, the seeds produced by haploid inducer lines show a wide range of abnormalities, and it seems important to document the phenotypic spectrum of the haploid and hybrid embryos/seeds produced in this particular cross to ensure the analysis is representative.*

Response: We agree with the reviewer’s comments. To address the reviewer’s concern, this revised version of our manuscript we have added a representative silique collected from each of the SeedGFP-HI × *lec1-1* and SeedGFP-HI × WT crosses as well as detailed description of the genetic crosses. Please see the revised Fig. 2b and legend. We conducted a large number of crosses; and in the revision we specified the number of siliques, seeds, and haploids examined in the figure legends of Fig. 2 and Supplementary Fig. 4.

*The phenotypic spectrum of mutant embryos/seeds produced by *lec1* or *lec1/+* plants should be documented side-by-side to enable a direct comparison.*

Response: We thank the reviewer for raising this issue. In this revised version, we have conducted the genetic crosses SeedGFPHI × WT as control and provided the results for direct comparison. Please see details in the revised Fig. 2b,c. We have also added the WT embryo and *lec1-1* embryo in the revised Fig. 2d; and provided WT or *lec1-1* plants and siliques in the Supplementary Fig. 4a-b.

*Finally: while I agree with the authors that the endosperm of seeds containing haploid (*lec1*) embryos is unlikely to have completely lost its wild type *LEC1* copy, this has not been shown.*

Response: We understand the reviewer's concern. In our HI × *lec1* experiments presented in the last version, it was very challenging to assess the genotype of LEC1 in the endosperms without harming the embryo before the maturation stage, as well as to distinguish the diploid and haploid embryos. In this revision, this particular concern was addressed by using the SeedGFP-HI line, which was generated by introducing an GFP marker driven by the seed storage protein 2S3 promoter (At2S3:GFP) into the original HI line. With this line, it was shown that the maternal genome in the endosperm at the late development stage of haploid seeds could be visualized by the presence of mottled GFP signals (Ravi, al., 2014). Thus, we obtained this line from Dr. Ravi for the manuscript revision. We have now repeated all HI experiments by using this SeedGFP-HI line. Indeed, we observed mottled GFP signals in some of the seeds from the crosses, suggesting that the endosperm of those seeds containing haploid embryos have the wild type alleles inherited from the maternal genome (in our case, LEC1). The new results are presented in details in the revised manuscript, please see Fig. 2b-e and lines 88-125.

Analysis of LEC1-GFP movement:

Fusions to multimeric GFPs are typically included in similar studies as non-mobile or less mobile controls. Such controls should be included here. Another control that may be considered is expression of LEC1 from an embryo-specific promoter, a promoter that does not confer expression in the endosperm. Expression of FUS3 with an MLI promoter has been reported to fully complement fus3 mutant embryos (Tsuchiya, 2004), and it would be interesting to see what activities an analogous LEC1 construct has.

Response: We thank the reviewer for this great suggestion. As suggested, we have generated the transgenic line: pPHE1::LEC1-GFP₃ *lec1-1* (PPL-GFP₃) as a non-mobile control. Our results show that the mobility of LEC1-GFP₃ was restricted to the endosperm only and no GFP signal was detected in the embryos. More importantly, the LEC1-GFP₃ transgene failed to rescue the *lec1-1* seed defects. Please see lines 175-182 in the revised manuscript and lines 274-277 in the revised Methods and Supplementary Fig.10.

At the same time, we have also generated the *pML1::LEC1-GFP* (PML) line for an exclusive expression of LEC1 in the embryo. Please see lines 183-188 and 288 - 290. As Expected, expression of the LEC1-GFP fusion in the PML was only observed in the embryos, and it could

successfully rescue the *lec1-1* seed defects (see Supplementary Fig. 11 for details). We would like to note that, based on the LEC1ChIP-seq data from Pelletier et al. (2017), LEC1 did not show any binding signal on the *AtML1* gene including its promoter region, suggesting that the activity of *AtML1* promoter is unlikely to be dependent on LEC1. This result is consistent with earlier observations that the *lec1* mutant endosperms do not show any obvious defect, and is also in line with our observations presented in this paper.

Figure 4: A reporter with a nuclear GFP would be a more appropriate control for showing that the PHERES promoter is not active in the embryo; the GFP of pMDC107 and hence PPg contains no nuclear localization signal (although, strangely, localization in c & f seems nuclear), and it seems problematic to compare cytoplasmic signals to the nuclear signals of PPL.

Response: We agree with the reviewer's suggestion to use a reporter with nuclear GFP as a control. In this revised manuscript, we have included the confocal images of GFP signals in the transgenic line: *pPHE1::ABI3-GFP*, as ABI3 is a nuclear localized protein. Please see the details in the revised manuscript in line 172-174 and Supplementary Fig. 9.

The figure legend does not specify which channels are shown in the various panels – from the looks of it, the rows showing “Globular”, “Heart”, “Bent” and “Maturation” embryos show only a single channel: cyan (autofluorescence) or green (GFP), which wouldn't make much sense.

Response: We thank the reviewer for flagging this. Each confocal image shown presents merged signals from GFP channel and PI channel in the revised Fig. 4a-u and specified in the now revised figure legend. We have provided a more detailed description in the revised Methods; see lines 322-325.

I don't understand why the PPg-m sample is used as a control in panel w. Detection by RT-PCR is likely much more sensitive than detection by fluorescence. A non-transgenic sample would be the appropriate control, showing the noise produced in the reaction; in addition, it would enable testing the expectation that the PHERES promoter does not confer expression in embryos.

Response: We appreciate the reviewer's concern. In the revised manuscript, RNA expression analysis of GFP in the wild type embryo (a non-transgenic line) was used as negative control. Please see the revised Fig. 4w.

Discussion:

It is well-documented that LEC1 is transcribed in the developing embryo until the maturation phase. Thus, the perhaps the most puzzling implication of this work is that LEC1 expression in the very domain that requires LEC1 activity cannot provide the activity. The authors should comment on this implication in their discussion.

Response: We agree that it appears puzzling, but it is also the most exciting aspect of our findings. Such an arrangement allows a spatial separation of the embryo and its key regulator and thus adds a new layer of regulation of embryo development. This would make sure that embryo development, including the maturation program, would not initiate until it is informed to do so, i.e., after successful fertilization, otherwise remains securely repressed during the rest of the plant's life cycle. Please see lines 234-244. To summarize the findings in this work and its implications, we have proposed an operating model for *LEC1* expression/accumulation in the endosperm and embryo in the revised discussion; please see Fig. 5.

REVIEWERS' COMMENTS

Reviewer #1 (Remarks to the Author):

The revised version of the manuscript answers my initial questions and interrogations. The additional figures, additional controls suggested by both reviewers and the last model figure enhance greatly the quality and ease of reading of the MS.

Only minor concerns:

On supplementary figure 7- a, c-PZL-5 is not described only c-PZL-2 in the legend. Are these independent transgenic lines? Please indicate in the legend c-PZL-2, 5, 7 and c-PLL1, 2, 5.

In the legend, the sentence "normal seeds with transgene and lec1-1 seeds without transgene" is not very clear, as they are all in lec1-/- except the WT. You should write [WT] seeds and [lec1] seeds instead, referring to the phenotypes.

Do you mean the transgene is segregating? I presume the mother plant was hemizygous for the transgene, which explains the 75% of N seed? As they are all in lec1 background, I would rather indicate that the lec1 seeds are restored or not to WT phenotype according to transgene's segregation, or use [].

On figure 5, the embryo is not "arrested", only the maturation step is skipped in lec mutants (no reserve, leaves instead cotyledons, no desiccation resistance...). Arrested refers to mutants with an altered morphology (division patterns, globular shape, ...), which is not really the case in lec mutants with all organs at the right place. This is well described in the plain text and I would change "arrested" by "lack of maturation"

Line 34: LEC1 is not a novel subunit of NF-YB type, it has been described long time ago. But it is certainly a different type of NY-YB with specific AA residues (D55 according to literature for example).

Reviewer #2 (Remarks to the Author):

The revised version of the manuscript includes a substantial amount of new data, including several controls that were added in response to the reviewer's comments. Most notably

- The description of crosses with fis2 & cdka;1 is now much more detailed, which should help non-specialist readers to better follow the narrative; in addition, key genotypes are verified by PCR (Fig. 1 and main text).
- A fluorescent marker was added to the haploid inducer line as a means of differentiating between diploid and haploid progeny (Fig. 2 and main text).
- A LEC1 protein tagged with a triple GFP (and thus too large for passive movement through plasmodesmata) was expressed in the endosperm to verify that LEC1 movement from the endosperm to into the embryo is required for normal embryo development (new Suppl. Fig. 10).
- LEC1 was expressed in the epidermis of developing embryos from the ML1 promoter, revealing that independent expression in the embryo is sufficient for LEC1 function (new Suppl. Fig. 11).

The last experiment, in particular, adds plausibility to the author's model (outlined in the new Fig. 5). A key prediction of this model is that LEC1 expression in the embryo is normally dependent on LEC1 protein provided to the embryo by the endosperm, which explains why embryos containing a wild type copy of the gene develop a mutant phenotype if their endosperm is mutant for LEC1. Indeed, independent expression of LEC1 in the embryo circumvents this block and restores normal development. (The discussion is potentially confusing on this point, saying that "LEC1 expression in the very domain that requires LEC1 activity cannot provide the activity": doesn't expression from the ML1 promoter show that activating the LEC1 gene in the embryo provide the needed activity?)

In my view, the revisions have much benefited the paper. The overall presentation is very clear; the experiments are quite original and were executed with much care; the resulting model is both surprising and compelling. Congratulations!

Minor comments:

Could "embryo-rescued" (page 2) be explained?

Is "pre-embryo" (page 8) in error for pro-embryo?

The term "un-rescued embryo phenotypes" (page 8) seems awkward.

Response to Reviewers' Comments

We are grateful to the reviewers for their insightful suggestions. Our detailed point-to-point responses to the reviewer's specific comments are presented below:

REVIEWER COMMENTS

Reviewer #1 (Remarks to the Author):

The revised version of the manuscript answers my initial questions and interrogations. The additional figures, additional controls suggested by both reviewers and the last model figure enhance greatly the quality and ease of reading of the MS.

Only minor concerns:

On supplementary figure 7- a, c-PZL-5 is not described only c-PZL-2 in the legend. Are these independent transgenic lines? Please indicate in the legend c-PZL-2, 5, 7 and c-PLL1, 2, 5.

In the legend, the sentence "normal seeds with transgene and lec1-1 seeds without transgene" is not very clear, as they are all in lec1-/- except the WT. You should write [WT] seeds and [lec1] seeds instead, refereing to the phenotypes.

Response: Thank you for pointing it out. Yes, c-PZL-2 and c-PZL-5 are independent transgenic lines. We have indicated this in the revised legend of supplemental figure 7. Thank you for your suggestion. We have revised the legend as suggested.

Do you mean the transgene is segregating? I presume the mother plant was hemizygous for the transgene, which explains the 75% of N seed? As they are all in lec1 background, I would rather indicate that the lec1 seeds are restored or not to WT phenotype according to transgene's segregation, or use [].

Response: Yes, the transgene is segregating and the mother plant was hemizygous for the transgene. We have used [] to indicate the seed phenotypes.

On figure 5, the embryo is not "arrested", only the maturation step is skipped in lec mutants (no reserve, leaves instead cotyledons, no desiccation resistance...). Arrested refers to mutants with an altered morphology (division patterns, globular shape, ...), which is not really the case in lec mutants with all organs at the right place. This is well described in the plain text and I would change "arrested" by "lack of maturation"

Response: Thank you for your suggestion. We have changed it to "lack of maturation" on Fig. 5.

Line 34: LEC1 is not a novel subunit of NF-YB type, it has been described long time ago. But it is certainly a different type of NY-YB with specific AA residues (D55 according to literature for example).

Response: Thank you. We have revised it to "nuclear factor Y (NF-Y) transcription factor LEAFY COTYLEDON1 (LEC1)". Please see line 36-37.

Reviewer #2 (Remarks to the Author):

The revised version of the manuscript includes a substantial amount of new data, including several controls that were added in response to the reviewer's comments. Most notably

- The description of crosses with *fis2* & *cdka;1* is now much more detailed, which should help non-specialist readers to better follow the narrative; in addition, key genotypes are verified by PCR (Fig. 1 and main text).*

- A fluorescent marker was added to the haploid inducer line as a means of differentiating between diploid and haploid progeny (Fig. 2 and main text).*

- A *LEC1* protein tagged with a triple GFP (and thus too large for passive movement through plasmodesmata) was expressed in the endosperm to verify that *LEC1* movement from the endosperm to into the embryo is required for normal embryo development (new Suppl. Fig. 10).*

- *LEC1* was expressed in the epidermis of developing embryos from the *ML1* promoter, revealing that independent expression in the embryo is sufficient for *LEC1* function (new Suppl. Fig. 11).*

*The last experiment, in particular, adds plausibility to the author's model (outlined in the new Fig. 5). A key prediction of this model is that *LEC1* expression in the embryo is normally dependent on *LEC1* protein provided to the embryo by the endosperm, which explains why embryos containing a wild type copy of the gene develop a mutant phenotype if their endosperm is mutant for *LEC1*. Indeed, independent expression of *LEC1* in the embryo circumvents this block and restores normal development. (The discussion is potentially confusing on this point, saying that “*LEC1* expression in the very domain that requires *LEC1* activity cannot provide the activity”: doesn't expression from the *ML1* promoter show that activating the *LEC1* gene in the embryo provide the needed activity?)*

Response: Thank you for your comments. To make it more straightforward and clear, we have changed the sentence from “...*LEC1* expression in the very domain that requires *LEC1* activity cannot provide the activity” to “Another puzzling question is why *LEC1* expression in the embryo is normally dependent on *LEC1* protein provided to the embryo by the endosperm.”, by adopting what you wrote – thank you.

In my view, the revisions have much benefited the paper. The overall presentation is very clear; the experiments are quite original and were executed with much care; the resulting model is both surprising and compelling. Congratulations!

Minor comments:

Could “embryo-rescued” (page 2) be explained?

Response: We have revised it to “the *lec1* homozygous plants rescued from embryo”. Please see line 42-43. The method for embryo rescue was described in the Method section. Please see line 301 -306.

Is “pre-embryo” (page 8) in error for pro-embryo?

Response: Thank you. We have revised it to “pro-embryo”.

The term “un-rescued embryo phenotypes” (page 8) seems awkward.

Response: We have revised it to “lec1 mutant”. Please see line 177.